# Robustness of Physiological Synchrony in Wearable Electrodermal Activity and Heart Rate as a Measure of Attentional Engagement to Movie Clips

**DOI:** 10.3390/s23063006

**Published:** 2023-03-10

**Authors:** Ivo V. Stuldreher, Jan B. F. van Erp, Anne-Marie Brouwer

**Affiliations:** 1Human Performance, Netherlands Organisation for Applied Scientific Research (TNO), 3769 DE Soesterberg, The Netherlands; 2Human Media Interaction, Faculty of Electrical Engineering, Mathematics and Computer Science, University of Twente, 7522 NB Enschede, The Netherlands; 3Human Machine Teaming, Netherlands Organisation for Applied Scientific Research (TNO), 3769 DE Soesterberg, The Netherlands; 4Artificial Intelligence, Faculty of Social Sciences, Radboud University, Thomas van Aquinostraat 4, 6525 GD Nijmegen, The Netherlands

**Keywords:** physiological synchrony, inter-subject correlations, attentional synchrony, EDA, electrodermal activity, heart rate

## Abstract

Individuals that pay attention to narrative stimuli show synchronized heart rate (HR) and electrodermal activity (EDA) responses. The degree to which this physiological synchrony occurs is related to attentional engagement. Factors that can influence attention, such as instructions, salience of the narrative stimulus and characteristics of the individual, affect physiological synchrony. The demonstrability of synchrony depends on the amount of data used in the analysis. We investigated how demonstrability of physiological synchrony varies with varying group size and stimulus duration. Thirty participants watched six 10 min movie clips while their HR and EDA were monitored using wearable sensors (Movisens EdaMove 4 and Wahoo Tickr, respectively). We calculated inter-subject correlations as a measure of synchrony. Group size and stimulus duration were varied by using data from subsets of the participants and movie clips in the analysis. We found that for HR, higher synchrony correlated significantly with the number of answers correct for questions about the movie, confirming that physiological synchrony is associated with attention. For both HR and EDA, with increasing amounts of data used, the percentage of participants with significant synchrony increased. Importantly, we found that it did not matter how the amount of data was increased. Increasing the group size or increasing the stimulus duration led to the same results. Initial comparisons with results from other studies suggest that our results do not only apply to our specific set of stimuli and participants. All in all, the current work can act as a guideline for future research, indicating the amount of data minimally needed for robust analysis of synchrony based on inter-subject correlations.

## 1. Introduction

Individuals that attend to narrative stimuli (e.g., movie clips or auditory narratives) show synchronized neurophysiological responses such as brain potentials and heart rate (HR) [1,2,3]. This is referred to as physiological synchrony. Stronger physiological synchrony among participants is generally related to higher degrees of shared attentional engagement [3,4,5,6,7]. Physiological synchrony is, for instance, higher among individuals who actively attend to a presented narrative compared to individuals who focus attention inward on a distracting task [1,3]. Additionally, when engagement decreases due to being presented with a narrative for the second time, physiological synchrony also decreases [4]. Physiological synchrony as marker of attentional engagement is even informative on the level of an individual within a group: the more the physiological responses of an individual synchronize with those of others attending to the narrative, the better this individual can recall the narrative [2,3,6].

Narrative-driven synchrony is most studied and most pronounced for neuroimaging modalities, such as the electroencephalogram (EEG), magnetoencephalogram (MEG) or functional magnetic resonance imaging (fMRI) [4,8,9]. Brain-to-brain synchrony is often quantified through inter-subject correlations [10]. Such inter-subject correlations amongst attending individuals are significantly higher than chance upon presentation of narrative stimuli [11], are reduced when individuals are distracted from the narrative [1], distinguish between individuals with different selective attentional focus to part of the presented stimulus [1] and are predictive of the occurrence of attentionally relevant stimuli in time [12]. In addition, brain-to-brain synchrony has been related with numerous behavioral metrics, such as stimulus retention, efficacy of advertising and efficacy of communication [2,5,7,13]. This relation between brain-to-brain synchronization, attention and behavioral outcomes has been established for both auditory and audiovisual narratives [6].

Narrative-driven synchrony has also been established in measures that reflect autonomic nervous system activity, such as HR and electrodermal activity (EDA) [2,3]. We refer to this as body-to-body synchrony. Although the brain is most closely involved in attentional processing, body measures can also index attention through reflecting arousal: the physiological state of activation of the body. Arousal and attention are closely related and share a common neural substrate [14]. HR and EDA are sensitive to changes in arousal and, thereby, are also associated with changes in attention. Indeed, HR and EDA respond to arousing, emotionally relevant events that are attentionally prioritized [15,16] and contribute to the monitoring of driver attention [17,18].

Similar to brain-to-brain synchrony, it has indeed been found that body-to-body synchrony reflects attentional engagement. Inter-subject correlations in such measures are significantly higher than chance upon presentation of narrative stimuli [3], are reduced when individuals are distracted from the narrative [3], distinguish between individuals with different selective attentional focus to part of the presented stimuli [2] and are predictive of the occurrence of attentionally relevant stimuli in time [12]. Although to a lesser degree than brain-to-brain synchrony, body-to-body synchrony has been related to some behavioral metrics, such as stimulus retention [2,3].

While body-to-body synchrony is associated with attentional engagement, it appears to be less robust and to a lesser degree related to attention than brain-to-brain synchrony. It has been argued that this may be so because body-to-body synchrony rather reflects arousal or emotional engagement, whereas brain-to-brain synchrony more directly reflects attention [2,19,20]. In addition, the potential information density of most brain measures strongly exceeds that of body measures in terms of modulation frequency and number of sensors. In general, the number of participants with significant inter-subject correlations (i.e., correlations exceeding chance-level expectations) was lower for HR than for EEG [21]. Inter-subject correlations in EDA and HR also distinguished less reliably between two selective attentional conditions than inter-subject correlations in EEG, though inter-subject correlations in HR were correlated with performance metrics in a similar way as inter-subject correlations in EEG [2].

In comparison with methods based on machine learning models, physiological synchrony is a potentially valuable method to implicitly monitor attention in real-life situations. Analyzing physiological synchrony allows the use of ecological stimuli (movies and narratives). Additionally, there is no need to train a model beforehand on previously collected (personal) data. While, as reviewed above, brain-to-brain synchrony seems more sensitive than body-to-body synchrony, the latter has advantages from the perspectives of user comfort and costs. The use of wearable EDA and HR sensors allows for the monitoring of physiological synchrony in real-world settings, such as classrooms or conferences [22,23]. The need to move to real-life settings has been stressed many times [24]. While physiological synchrony as determined by using wearables seems very suitable for real-life research and applications, questions regarding the analysis of physiological synchrony remain. The computational approach used to quantify physiological synchrony may affect potential outcomes. Algumaei, for instance, reported that physiological synchrony in the electrocardiogram (ECG) had predictive value regarding team performance using a multidimensional recurrence quantification analysis, but not using dyadic linear cross correlations [25]. Linear cross correlations, on the other hand, are successfully employed to assess the attentional engagement towards narrative stimuli [2,3]. In settings where individuals are presented with the same narrative stimulus, nonlinear analyses may not be required as there are no asymmetric relationships to be captured. More complex nonlinear analysis may also require more data to be collected to set additional modeling parameters, such that inter-subject correlations are the more suitable approach. 

However, for such inter-subject correlations, the requirements in terms of amount of data needed to obtain robust inter-subject correlations also remains unclear. There are still few studies employing inter-subject correlations in HR and EDA and reporting requirements in terms of data used. Pérez et al. [3] reported that, for single 60 s movie clips, only a few participants showed significant inter-subject correlations, but when aggregating over all 16 movie clips used in their study, the majority of participants showed significant inter-subject correlations. Stuldreher et al. [2] found that when selecting parts of an entire 66 min audio stimulus, generally, less participants could be classified in the correct attentional condition. Though these results indicate that the significance of inter-subject correlations depends on the amount of data used, it remains unclear what the specific relationship is between the amount of data and inter-subject correlations. There are no criteria for minimum group size or minimum stimulus length for sensible physiological synchrony results. In addition, it is unclear whether it is wise to increase the amount of data through recording responses from more people, or longer stimuli, if the option is there.

Here, we explore the amount of data required to obtain robust results for body-to-body synchrony (HR and EDA) recorded with wearable equipment while watching movies. The amount of data is varied by the number of participants and the duration of (audiovisual) stimuli. In addition, we varied attentional instruction such that participants were either asked to attend to the movie, or to respond to a certain visual cue. As described later, this manipulation did not seem to affect any outcome measure and is not the focus of the paper, but is discussed at the end.

## 2. Materials and Methods

### 2.1. Participants

Thirty participants (14 female), between 19 and 64 years old, with an average age of 39.4 years and a standard deviation of 15.8 years, were recruited through the institute’s participant pool. Before performing the study, approval was obtained from the TNO Institutional Review Board (IRB). The approval is registered under reference 2020-117. All participants signed informed consent before participating in the experiment, in accordance with the Declaration of Helsinki. After successful participation, participants received a small monetary compensation for their time and travel costs. 

### 2.2. Materials

EDA and HR were recorded using wearable systems. EDA was recorded using EdaMove 4 (Movisens GmbH, Karlsruhe, Germany) worn at the wrist of the non-dominant hand. The EdaMove 4 uses two solid gelled Ag/AgCl electrodes (MTG IMIELLA electrode, MTG Medizintechnik, Lugau, Germany, W55 SG, textured fleece electrodes, 55 mm diameter) recording signals from the palmar surface of the hand. A constant direct current (DC) voltage of 0.5 V was applied to the skin. Measurements were conducted at a sampling rate of 32 Hz with an input range of 2–100 μS and with a resolution of 14 bits. HR was recorded using the commercially available and sport-oriented Tickr chest strap (Wahoo Fitness, Atlanta, GA, USA). The device reports HR in beats per minute (bpm), at a rate of one value per second and a resolution of one bpm. Data were streamed over Bluetooth to a smartphone for local saving on the device through the Wahoo Fitness application (Version 1.36.0.291). Both EdaMove 4 and Tickr have been demonstrated before to provide signal quality close to high-end lab equipment [26], and to be suitable for the measurement of attention-modulated physiological synchrony [27].

### 2.3. Design

Participants performed the experiment one by one. All participants were presented with six movie clips of approximately 10 min duration (09:48 ± 00:41 min). The details and URLs can be found in Table 1. The movie clips were selected from the Dutch YouTube channels NPO3 and KORT! and featured 10 min stories. The presentation order was randomized across participants. We chose these clips as we are not aware of any affective movie databases containing six approximately 10 min videos of neutral valence and moderate arousal, preferably in Dutch. Although the clips are not standardized in terms of valence and arousal, the movies are comparable in terms of length and type. All contain an emotional narrative that develops throughout the 10 min video, as judged at face value by two of the authors. We avoided films with strongly arousing content, such as the death of characters, physical violence and verbal violence. The films are of the arthouse genre.

Besides the robustness of inter-subject correlations as a function of measurement duration and group size, we also aimed to study the effect of a double task on inter-subject correlations. For this, every 2–10 s, a millisecond counter (red font) was displayed in the center of the screen on top of the movie. This counter disappeared after 5 s or upon a button press of the participant. Alternating between movie clips, participants were instructed to attend to the movie clip and to ignore the counter or to respond as quickly as possible to the appearance of the counter by pressing the spacebar (i.e., performing the psychomotor vigilance task (PVT); [28]). These conditions are referred to as movie attending (MA) and task attending (TA), respectively. In the TA condition, PVT performance was determined by the mean reaction time after appearance of the counter.

To gain insight into the effect of the attentional instructions on the selective attentional performance of participants and to relate attentional performance to physiological synchrony, participants in both conditions were asked to answer 10 questions about the narrative of the movie clips immediately after each movie clip. These questions and the correct answers (in Dutch) can be found in the Appendix A. 

### 2.4. Analysis

#### 2.4.1. Pre-Processing

All data and scripts are available on https://github.com/ivostuldreher/robustness-of-physiological-synchrony (accessed on 28 December 2022). Data processing was performed using MATLAB 2021a software (Mathworks, Natick, MA, USA). Both EDA and HR were time-locked to the onset of the movie clips using locally saved timestamps. Before starting the experiment, we synchronized device clocks by using the same online clock on all devices. EDA was imported using proprietary scripts using the unisens4matlab toolbox. We followed the pre-processing procedure of recent work using the same sensor [29]. Periods of signal loss due to loose electrodes were identified and marked for removal later on in the process. The signal was filtered using a 3 s Savitzky–Golay filter to overcome the quantization noise in the signal. The fast-changing phasic component and slowly changing tonic component of the EDA were then separated using Continuous Decomposition Analysis as implemented in the Ledalab toolbox for MATLAB [30]. In further analysis, we used the phasic component of EDA, as this component is mainly related to responses to external stimuli. Regarding EDA, from this point, we refer to the phasic component of EDA. After decomposition of the signal, we removed those parts of the signal that were marked for removal earlier on. If more than 30% of data recorded from an individual were marked for removal, the entire recording was removed for further analysis. Based on this criterion, data of three out of the thirty participants were removed from further analysis. From three additional participants, a portion of the EDA data were removed as they contained periods marked as artifactual. From these participants, 0.8%, 6.0% and 10.8% of data were removed. In total, we used datasets of 27 individuals in further analysis, of which, for three participants, 0.8%, 6.0% and 10.8% of the EDA data were removed.

The HR data were exported from the Wahoo Fitness application in a .fit format and imported in MATLAB using proprietary scripts based on FIT SDK 21.38.00 [31]. Data of one participant were lost due to a failed recording. Suspicious samples in the data were removed based on two criteria: (1) samples higher than 200 bpm or lower than 30 bpm because we consider such values unrealistic in the current setting and (2) samples more than 25% different from the value 1 s before; such changes in HR are considered unrealistic [32]. This was only true for one participant, for whom 0.3% of data were removed. Data of participants where samples equal to the previous sample are 50 times more prevalent than samples different from the previous sample were also removed from further analyses, as they indicate a malfunctioning HR sensor. This affected six participants. HR data of three participants were completely removed from further analyses; for the three other participants, data recorded during two of the six movie clips were removed. In sum, the HR data of 26 participants were used in further analysis, of which, for three participants, data of two out of six movie clips were discarded. 

EDA and HR data were then epoched and time-locked to the start of each movie clip as further analyses were conducted on the physiological data recorded during the movie presentation. Data were time-locked to the movie based on the computer time at the start of each movie presentation, and were saved to a .csv file and cut to the duration of each movie. 

#### 2.4.2. Inter-Subject Correlations

We computed physiological synchrony between participants during each movie clip using inter-subject correlations, following methods used in earlier work [2,3,12]. Dyadic inter-subject correlations were computed for all unique dyads. Inter-subject correlations were computed in 15 s windows sliding at 1 s increments over the entire epoch of interest (here, the entire movie clip), and were subsequently averaged across the entire epoch. Thus, for each physiological measure and each epoch, we obtained an N×N matrix of inter-subject correlations, where N represents the number of participants. For each participant, participant-to-group inter-subject correlations were computed by averaging over all values in a row, excluding the diagonal. 

To investigate whether the movies and attentional instruction affected inter-subject correlation, a two-way ANOVA with the independent variables movie and attentional condition was performed.

#### 2.4.3. Circular Shuffle-Based Significance Test

To test the significance of participant-to-group inter-subject correlation values, we used the circular shuffle statistic, following a previous work [3]. Each participant’s physiological signal was circular shifted by a random amount within the epoch length. The inter-subject correlations and participant-to-group inter-subject correlations were then computed with these circular shuffled data. This procedure was repeated 50 times for each participant. Statistical significance of the inter-subject correlations was assessed using a one-sided independent sample *t*-test comparing the actual correlations to those resulting from the shuffled data (significance threshold p<0.05).

#### 2.4.4. Effect of Stimulus Duration and Group Size on Inter-Subject Correlation Significance

Our aim was to investigate how stimulus duration and group size affect the inter-subject correlation significance. 

To investigate the effect of stimulus duration, we artificially created shorter and longer movie clips. The first step was to combine the six 10 min movie clips into 60 min movie clips. A total of 720 different orders of combinations of 10 min movie clips are theoretically possible, i.e., six factorial. Due to computational constraints, we chose to combine the movie clips in six different orders based on the following Latin square, where each row corresponds to one of the six orders used:$$\begin{array}{cccccc} 1 & 2 & 3 & 4 & 5 & 6\\ 2 & 5 & 4 & 6 & 1 & 3\\ 5 & 1 & 6 & 3 & 2 & 4\\ 3 & 4 & 5 & 1 & 6 & 2\\ 6 & 3 & 2 & 5 & 4 & 1\\ 4 & 6 & 1 & 2 & 3 & 5 \end{array}$$

The second step was to only select the first x minutes of data, where we varied x: 30 s, 1 min, then in 2 min increments up to 19 min and then from 24 min in 4 min increments to 60 min. For each subsegment of data, we performed the abovementioned circular shuffle analysis to investigate the statistical significance of the inter-subject correlations. 

To investigate the effect of group size on inter-subject correlations, we varied the group size from two to twenty-seven participants, at one-participant increments. As there are many subsets of participants possible, for each group size, we selected 50 random subsets of participants using the ”randsample” function as implemented in MATLAB. For each subset of participants, we investigated (for all of the above subsegments of data) which fraction of participants showed significant synchrony.

## 3. Results

### 3.1. Effect of Movie and Condition on Inter-Subject Correlation

To investigate whether the movies and attentional instruction affected inter-subject correlation, a two-way ANOVA with the independent variables movie and attentional condition was performed. There was no effect of attentional instruction on inter-subject correlations in HR ($F\left(1,5\right)<10^{-4}$, p=0.979) and EDA (F(1,5)=0.15, p=0.698). While there was a main effect of movie on inter-subject correlations, both for HR (F(1,5)=7.57, p<0.001) and EDA (F(1,5)=3.79, p≤0.003), there was no interaction with attentional instruction (HR: F(1,5)=0.88, p=0.498; EDA: F(1,5)=1.25, p=0.286). Therefore, in the remaining analyses, we collapse over the two conditions.

### 3.2. Significance of Inter-Subject Correlations

We first investigated the significance of inter-subject correlations in response to each of the movie clips separately. Figure 1 depicts the HR and EDA participant-to-group inter-subject correlations for each participant and each movie clip compared to a circular shuffle-based chance-level distribution. The figure illustrates the effect of movie clips on inter-subject correlation as described in the previous section. The majority of participants show significant HR inter-subject correlations in response to the movies “Chauffer”, “De Chinese Muur” and “One of the Boys”, while this is not the case for “El Mourrabi”, “Samuel” and “Turn it Around”. Inter-subject correlations in EDA show a similar pattern.

### 3.3. Dependency of Significant Inter-Subject Correlation on Stimulus Duration and Group Size

We then investigated how the percentage of participants with significant inter-subject correlations varied with stimulus duration and group size. Figure 2 shows how the percentage of participants with significant inter-subject correlations varies with stimulus duration and group size, for HR (top plots) and EDA (bottom plots), when averaged over the six movie orders and the fifty random subsets of participants. The left plots depict how the percentage varies with stimulus duration, for different group sizes. The right plots depict how the percentage varies with group size, for different stimulus durations. Additionally, Figure 3 shows the standard deviation of this percentage of participants with significant inter-subject correlations across the different movie orders as a function of stimulus duration in the left plots and across the fifty random subsets of participants as a function of group size in the right plots.

In Figure 2, the left plots show a generally increasing percentage of participants with significant inter-subject correlations with increasing stimulus duration, for all group sizes and for both EDA and HR. A small exception occurs in EDA when including only the first 30 s or one minute of data in the analysis; the percentage of participants with significant inter-subject correlations is actually higher for these stimulus durations than when including two to four minutes of data. The left plots in Figure 3 show that the percentage of participants with significant inter-subject correlations is strongly affected by the specific movie clip, depicted by the relatively large standard deviations for short stimulus durations (including only a single movie clip). 

The right plots in Figure 2 show a generally increasing percentage of participants with significant inter-subject correlations with increasing group size, for all stimulus durations and for both EDA and HR. We observe that a larger group size leads to less dependence on the specific sample of participants, indicated by the decreasing standard deviation for increasing group size as depicted in the right plots of Figure 3.

### 3.4. Comparing Effects of Stimulus Duration and Group Size on Significant Inter-Subject Correlation

The previous section describes two methods of increasing the total amount of data used (more movies—stimulus duration, and more participants—group size) and shows the expected pattern that more data lead to a higher percentage of participants with significant inter-subject correlations. This raises the question of whether it is wise to increase the number of participants or the length of the stimulus if the option is there. Therefore, we expressed both methods of varying the amount of data in terms of total duration of included data. For instance, a group size of 10 participants and a stimulus duration of 40 min lead to a total duration of included data of 400 min, and a group size of 20 participants and a stimulus duration of 20 min also leads to a total duration of included data of 400 min. Figure 4 shows the effects of both stimulus duration and group size in terms of total amount of data included, expressed in minutes for HR (top) and EDA (bottom). Each dot in the plot refers to the average percentage of participants with significant inter-subject correlations for a specific stimulus duration and group size. The color of the dots varies with group size; the size of the dots varies with stimulus duration. The shaded areas depict the standard deviation across the fifty random subsets of participants and six movie orders. The dots close in total amount of data included are also close in the percentage of participants with significant inter-subject correlations; dots are all on the same curve. The graphs, thus, show that while increasing the amount of data increases the percentage of participants showing significant inter-subject correlations, it does not seem to matter whether the amount of data is increased by increasing the stimulus duration or the number of participants.

### 3.5. Correlation between Performance Measures and Inter-Subject Correlation

Last, we investigated whether the inter-subject correlations correlate with performance on questions about the content of the movie clips. Correlations between inter-subject correlations and performance on questions about the content of the movie clips are non-significant, as indicated in Table 2. However, when collapsing over all movies, there is a significant correlation between HR participant-to-group synchrony and number of correct movie questions (last row in Table 2).

## 4. Discussion and Conclusion

### 4.1. Significance of Inter-Subject Correlations for Different Stimuli

In the current work, we investigated the robustness of physiological synchrony in EDA and HR when presented with narrative stimuli. This is of interest because physiological synchrony can reflect shared attentional engagement. A first premise for a robust relation between physiological synchrony (assessed through inter-subject correlations) and attention is significant inter-subject correlations, that is, inter-subject correlations that are higher than one would expect based on chance. We found that inter-subject correlations depend on the specific movie stimulus. For some movies, 78% of participants show significant participant-to-group inter-subject correlations, while for others, only 30% of participants show significant inter-subject correlations. We do not know what movie characteristics cause this difference. It may be that some movies were more engaging than others. It may also be that low-level characteristics drawing attention in a bottom-up fashion were more prevalent in some movies than in others. Inter-subject correlations are known to be affected by low-level characteristics of movies that draw attention in a bottom-up fashion [12,19,20]. Previously, we found that moments of high synchrony when listening to an audiobook did not correspond with scenes identified as overall ”attentionally engaging” by an independent group of listeners. Instead, we suggested that moments of high synchrony corresponded with relatively low-level engaging moments, such as swear words or salient intonation [12]. With the use of EEG, it has been reported that moments of high inter-subject correlations corresponded to short suspenseful moments in the movie [4,11]. Though the present study did not show an effect of attentional task (either performing the PVT or not), several studies, including [2], demonstrate the effect of top-down guided attention on physiology. Such findings refute the idea that physiological synchrony is completely caused by involuntary, bottom-up drawn attention. However, the exact relation between inter-subject correlations in HR and EDA and experienced attentional engagement is not clear yet.

### 4.2. Significance of Inter-Subject Correlations with Varying Amounts of Data Included

Our main question was how the prevalence of significant inter-subject correlations depends on the duration of the stimulus and on the size of the participant group. As expected, with increasing stimulus duration and increasing group size, the percentage of participants that show significant inter-subject correlations also increases. When using data of all participants, with stimulus durations of up to ten minutes, the percentage of significant inter-subject correlations is movie-dependent and low on average (<50%). Aggregating over two or three ten-minute movies already results in fairly robust significance levels, above 80%. Similarly, when using all 60 min of stimuli, for small group sizes (up to 10 participants), the percentage of significant inter-subject correlations is dependent on the specific sample of participants and low on average. With larger group sizes, the fraction of significant inter-subject correlations is high on average (>85%) and less dependent on the specific sample. 

It appears that it does not matter in which way the total amount of data is reached. Robust inter-subject correlations (80% of participants with significant inter-subject correlations) are reached with a total amount of data of roughly 10 h for both HR and EDA. It does not matter whether this total amount of data is reached by increasing the number of participants or by increasing the stimulus duration. Note that for small group sizes, the percentage of participants with significant synchrony strongly depends on the specific sample of participants (see Figure 3). We, therefore, suggest that researchers conduct similar studies to include at least 10 participants.

A small exception occurs in EDA when including only the first 30 s or the first minute of each movie stimulus in the analysis. The significance of inter-subject correlations is actually higher when including only the first minute of data than when including the first two up to the first four minutes of data (Figure 2, bottom left panel). When examining how EDA inter-subject correlations change over the course of the movies (see Appendix A), we observe high correlation values in the first 60 s of the each of the six movies. The figures in [19] show a similar pattern, with relatively high inter-subject correlations in the first 60 s of the stimulus. We think this is related to the very high levels of EDA at the beginning of each stimulus movie, that are necessarily followed by a sharp drop (see bottom panel in Appendix A). Such EDA patterns are commonly found in studies displaying stimuli to observers, and are presumably related to the arousal associated with the presentation of ”something new”. We also note that HR shows a similar, albeit more modest, pattern of high values at the start of each movie, followed by a drop (top panel in Appendix A). We think arousal and attentional engagement are closely linked. For the special case of the start of a stimulus, high levels of EDA, and high inter-subject correlations, may reflect general stimulus-driven arousal and engagement, which is somewhat apart from attentional engagement with the stimulus’ content. 

We cannot be sure whether our results are generalizable to similar studies or are a finding specific for our sample of participants and videos. There is a very limited number of studies assessing inter-subject correlations in EDA or HR of observers, let alone reporting their significance. However, two recent studies assessing significance of inter-subject correlations in HR present results similar to ours. Pérez et al. [3] report significance of inter-subject correlations in HR of 27 participants presented with 16 one-minute audiobook fragments. When aggregating over all audiobook fragments, thus including 27×16=432 min of data in total, 63% of participants show significant inter-subject correlations. This point coincides with the curve in Figure 4, indicating the results presented here may be comparable to results found using other stimuli. Madsen and Parra [21] obtain significant inter-subject correlations in HR for 66% participants, while using a total of 920 min of data recorded during the viewing of instructional videos. This reported prevalence of significance is on the lower end compared to our average results, but is still higher than the minimum percentage of participants with significant inter-subject correlations that we found for the same amount of data. These results suggest that the reported relation between the amount of included data and inter-subject correlation significance applies to data beyond our specific set.

A premise for inter-subject correlations when presented with narrative stimuli is that individuals process narrative stimuli comparably [21]. While our sample of participants varied quite widely in age and contained about equal numbers of males and females, samples of participants that contain even more differing individuals may start to violate the premise of comparable processing. For instance, individuals with autism spectrum disorders (ASD), depression or first-episode psychosis are known to show more varying neural patterns and, thus, reduced neural inter-subject correlations during naturalistic stimulus presentations [33,34,35,36]. When measuring inter-subject correlations across a group of individuals, among which there are individuals with non-typical neural patterns, we expect that adding more data by including longer stimuli will result in a lower fraction of participants with significant inter-subject correlations than when adding more data by including more individuals with typical neural patterns. Conversely, when measuring inter-subject correlations across a group of ”typical” individuals, adding more data by using a longer stimulus is expected to lead to a higher fraction of participants that show significant synchrony than when adding ”non-typical” participants. 

### 4.3. Inter-Subject Correlations as Measure of Attentional Engagement

For validating inter-subject correlation as a measure of attentional engagement, and with an eye on the potential applications of this marker, a demonstrated relation between inter-subject correlation and behavioral, attentional outcomes is important. For individual movie clips, we did not find significant correlations between participant-to-group inter-subject correlations and number of correctly answered movie questions in this study. When aggregating over all six movie clips, the correlation was significant for HR (r=0.20, p=0.010), though not for EDA (r=0.09, p=0.234). These results are in line with higher proportions of participants showing significant inter-subject correlation for HR than for EDA. It also indicates that for a robust relation between inter-subject correlations and attention, sufficient amounts of data are needed. 

The finding presented here is consistent with our previous work, in which inter-subject correlations in HR were predictive of selective attentional performance, but inter-subject correlations in EDA were not [2]. The correlations between inter-subject correlations and number of correctly answered questions presented here are low (r=0.20 when aggregated across movies). Inter-subject correlations, thus, only explain a small part of the variance in the answers to questions about the content of the video. Note that the strength of the relation between attentional engagement (as estimated through inter-subject correlation) and performance strongly depends on the sensitivity of the behavioral measures. E.g., when questions that are too easy or too hard are used, an association is not found. Furthermore, whereas answers to questions on the movie contain information on content-level attention, inter-subject correlations most likely also capture more low-level attentional characteristics, driven among others by stimulus saliency [12] and emotional engagement [19,20].

### 4.4. Is Higher Inter-Subject Correlation Always Better?

In the current work, we tested how participant group size and stimulus duration influence inter-subject correlations and their significance, with the goal of maximizing the fraction of participants with significant inter-subject correlations. This does not mean that higher inter-subject correlation is always better from the point of view of its potential value as a measure of attentional engagement. In fact, in the theoretical case that physiological signals of all participants are perfectly synchronized, the inter-subject correlations would lose their value since there is no physiological variability to relate to behavioral variability [37]. However, if the inter-subject correlations do not exceed the chance level because of a lack of data, any differences in attentional engagement due to an intervention or personal differences are also not visible. An important question is where the sweet spot lies wherein a piece of stimulus evokes just enough synchrony to obtain significant inter-subject correlations, but not enough to saturate the individual signals of interest [38]. This sweet spot may also be context-dependent. For instance, when evaluating how effective a certain stimulus is in attracting attention compared to a second stimulus, the inter-individual differences are of less interest than when investigating which group of participants shows the most engagement with a certain stimulus.

### 4.5. Conclusions

This study explored how the amount of data included in analyses influences the fraction of participants that show significant inter-subject correlations and showed that the source of the data (i.e., more participants or longer stimuli) is irrelevant. Future research should investigate the generalizability of this finding over different sets of stimuli and participants. We hope our results can help guide future researchers when setting up studies on narrative-driven inter-subject correlations. Through our work, we hope to contribute to the shift in research from controlled laboratory settings and high-end equipment to real-life settings and wearable sensors. 

## Figures and Tables

**Figure 1 sensors-23-03006-f001:**
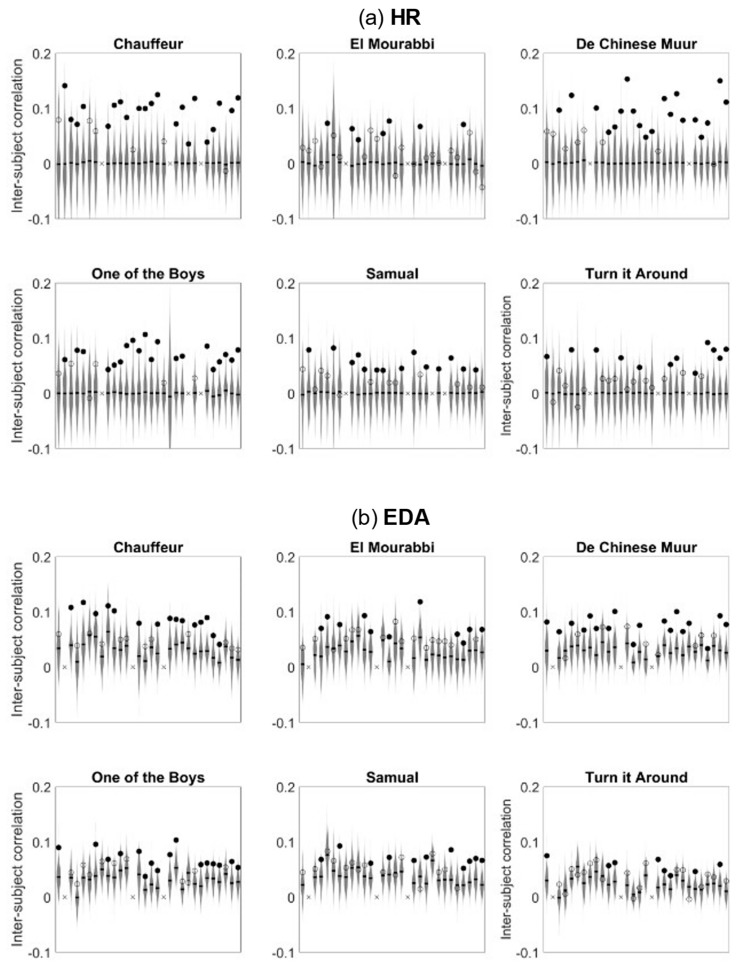
Participant-to-group physiological synchrony for each participant’s HR (**a**) and EDA (**b**). In each window, each marker refers to a participant. Filled markers depict inter-subject correlations exceeding chance-level correlations based on 500 trials of circular shuffle (depicted by the grey distributions); open markers depict inter-subject correlations not exceeding chance level. x depicts missing data.

**Figure 2 sensors-23-03006-f002:**
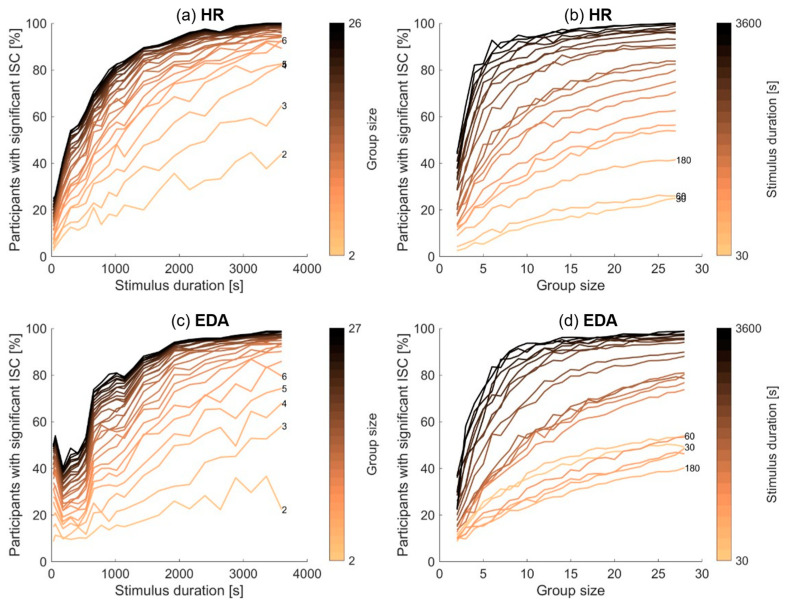
Percentage of participants with significant inter-subject correlations for HR (**top**; **a**,**b**) and EDA (**bottom**; **c**,**d**) as a function of stimulus duration (**left**; **a**,**c**) and participant group size (**right**; **b**,**d**), averaged over the six movie orders and 50 subsets of participant combinations. The color of the lines refers to the group size in the left plots and to the stimulus duration in the right plots, as do the numbers on the right side of some of the lines.

**Figure 3 sensors-23-03006-f003:**
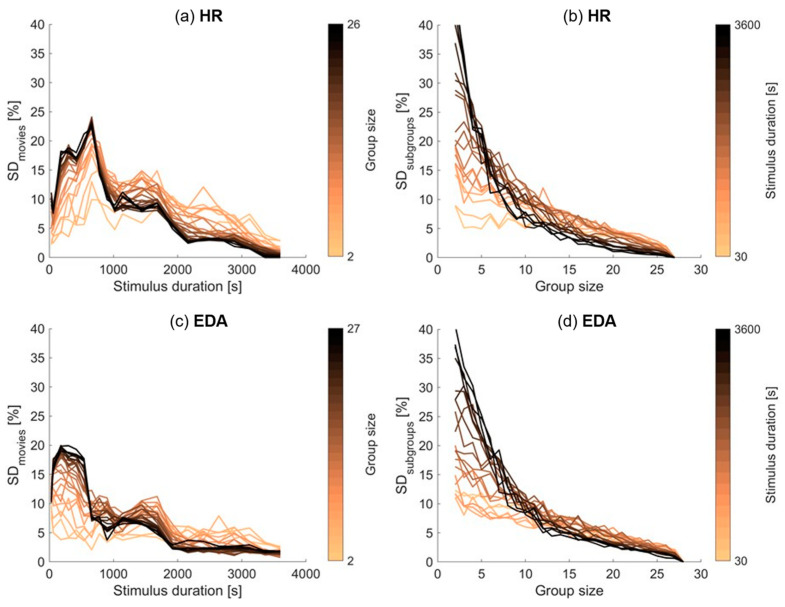
Standard deviation across movies (**left a**,**c**) and subgroups (**right**; **b**,**d**) of the percentage of participants with significant inter-subject correlations for HR (**top**); (**a**,**b**) and EDA (**bottom**; **c**,**d**) as a function of stimulus duration (**left**; **a**,**c**) and participant group size (**right**; **b**,**d**). The color of the lines refers to the group size in the left plots and to the stimulus duration in the right plots.

**Figure 4 sensors-23-03006-f004:**
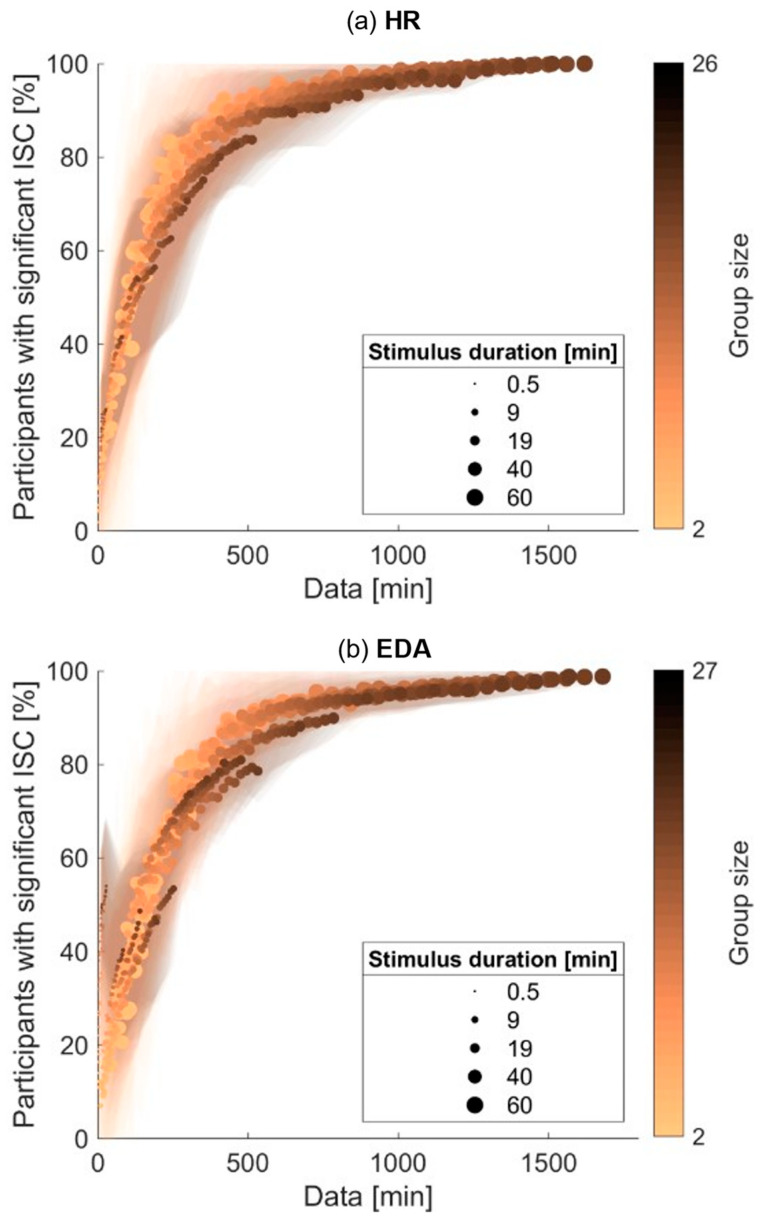
Fraction of participants with significant inter-subject correlations for HR (**a**) and EDA (**b**) as a function of the total minutes of data included in analysis, varied through varying stimulus duration and group size. For each datapoint, the stimulus duration is reflected by the marker size and the group size is reflected by the marker color.

**Table 1 sensors-23-03006-t001:** Description of shown movie clips, categorized by name, duration and URL.

Name	Duration	URL (Accessed on 30 November 2020)
Chauffeur	09:45	https://www.youtube.com/watch?v=jaFmvyH7dW8
El Mourabbi	09:04	https://www.youtube.com/watch?v=X9bJou2gKxo
De Chinese Muur	09:50	https://www.youtube.com/watch?v=yjGFuhPy3Qo
One of the Boys	10:58	https://www.youtube.com/watch?v=PsGAuhgQ97k
Samual	09:45	https://www.youtube.com/watch?v=VUseoqCVnj4
Turn it Around	09:26	https://www.youtube.com/watch?v=beC7IpQpTz4

**Table 2 sensors-23-03006-t002:** Spearman correlations between number of correct answers on questions about the content of the movie and participant-to-group physiological synchrony.

Movie	HR	EDA
Chauffeur	r=0.13, p=0.522	r=0.02, p=0.905
El Mourabbi	r=0.14, p=0.502	r=0.14, p=0.490
De Chinese Muur	r=0.27, p=0.171	r=0.12, p=.0541
One of the Boys	r=0.12, p=0.550	r=0.18, p=0.341
Samual	r=0.06, p=0.751	r=0.12, p=0.571
Turn it Around	r=0.20, p=0.320	r=−0.08, p=0.659
Overall	r=0.20, p=0.010	r=0.09, p=0.234

## Data Availability

The MATLAB scripts and data reproducing the results in this study are publicly available on https://github.com/ivostuldreher/robustness-of-physiological-synchrony (accessed on 28 December 2022).

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
