# Peer review of "Robustness of Physiological Synchrony in Wearable Electrodermal Activity and Heart Rate as a Measure of Attentional Engagement to Movie Clips"

_sensors, 2023, doi:10.3390/s23063006_

Round 1

Reviewer 2 Report

This paper collected electrodermal activity and heart rate signal in presence of some audio visual stimuli and compared them across the subjects with different window lengths. The idea of this paper is relatively old, and the contribution of this paper is not clear which makes it not suitable for publication at the current version. My comments are as follows.

1.       After reading the paper the contribution of this paper is not clear to me. It’s obvious that there will be correlation in EDA/HR within the subjects under a specific environment and specific scenario. So, the authors need to demonstrate what’s new in there study and why is it significant.

2.       The authors should work on improving the readability of this paper. Often time it is difficult to understand what message they want to convey, and sometimes big and lengthy sentences make it worse.

3.       A brief explanation of electrodermal activity and HR and how they are related to attentional engagement should be presented with proper reference.

4.       The paper is missing proper literature review. The authors should cite more related and recent literatures regarding the use of EDA and HR.

Reviewer 3 Report

Manuscript ID: sensors-2158980

Title: Robustness of physiological synchrony in wearable electrodermal activity and heart rate as a measure of attentional engagement to movie clips

Recommendation: Major revision

Brief summary

This manuscript investigates how the demonstrability of physiological synchrony between heart rate (HR) and electrodermal activity (EDA) varies with group size and stimulus duration. In particular, synchrony was quantified through inter-subject correlations, considering a test population of 29 subjects. The participants were made watch six 10-minute movie clips; the results show that synchrony improves with data quantity, provided either with a wider sample size or with a higher test duration.

Broad comments

The topic is relevant, since the study focuses on the use of non-intrusive wearable sensors to acquire the signals of interest, resulting in a practical method for the assessment of attentional engagement.

The writing of the manuscript should be improved, since the English is often not so fluent. Moreover, there are typos and incomplete sentences throughout the paper that should be revised. It seems that the authors have not re-read at all the manuscript before submitting it to Sensors journal. Thoroughly re-reading the papers could help authors to correct these issues and improve readability (e.g. through the use of short and clearer sentences). A particular attention should be paid to the use of punctuation.

The contribution with respect to the state of the art should be better evidenced in the introduction section of the paper. Also, the value of the manuscript in the context of Sensors journal should be better considered, since at present no significant evidence in this sense can be easily perceived from the paper.

The article should be better contextualized in the literature background; some references in the domain of Sensors could be added, also to highlight the work steps forward with respect to the state of the art.

Some suggestions are provided in the next comments, which may help the authors in improving the quality of this paper.

Specific comments

Keywords: the authors mention EEG; however, in the abstract only HR and EDA signals are reported. Also, this study seems to be focused in signals different from EEG. Please check.

Abstract: the authors should briefly report how they measured HR and EDA. Also, no mention to wearable sensors is made, whereas the title report a reference to “wearable electrodermal activity and heart rate”.

Furthermore, the results should be reported in a clearer way, since the readability of the related part in the abstract is not optimal. Moreover, only results for HR are reported and not for EDA (whereas on the contrary in the title is the only signal to be mentioned). Please check.

Introduction: this section is difficult to read and clearly understand. The readability should be improved; moreover, the authors should at first report the state of the art, then clearly state the aim of the paper, without blending the two of them. Please revise.

Lines 35-37: the authors should better explain this concept, also reporting the main findings of the cited references.

Lines 44-49: this sentence should be checked and rephrased for the sake of readability. The same for lines 55-60.

Lines 97-98: a non-complete sentence seems to be present. Please check.

Lines 98, 101: the references format is different from the previous citations. Please check.

Lines 109-110: they are the same as lines 96-97. Please check.

Line 119: please check, there could be a typo (results are not reported in this section!).

Lines 125-138: what about the measurement uncertainty of the employed sensors?

Lines 143-144: how can the authors classify the level of engagement of the clips? Why did not they use labelled databases, with movies classified in terms of valence and arousal, for example with Self Assessment Manikins from IAPS?

Line 152: please spell out the PVT acronym when used for the first time.

Line 161: there is an error in the cross reference. Please check. The same for lines 269, 275, 285, 291, 300, 305, 337, 387, 389, 394, 398. Please check.

Lines 180-183: this part should be moved to the Results section. The same for lines 192-194.

Lines 187-191: these choices should be motivated.

Lines 216-219: the p-value considered for statistical significance should be reported.

Figure 1: this numbering is repeated for two figures (consequently, the following figures numbering is wrong), please check. Furthermore, the subplots should be clearly identifiable (e.g. with letters). Moreover, the axes labels should be always reported, otherwise the readability is not good. Please note that each figure should be referred within the text and properly discussed.

Lines 311-312: physiological inter-subject variability has a different weight in the two options. The authors should consider this aspect and discuss it, at least in the discussion section.

Table 2: it should be referenced within the text and briefly discussed.

Section 4: it should be titled “Discussion and conclusion”, otherwise the manuscript lacks a conclusion section.

Lines 349-353: this is a weakness of the work, since the authors did not adopt a database already classified in terms of subjects’ emotional responses. Again, this choice should be motivated.

Lines 369-379: quantitative results should be reported, for the sake of readability.

Lines 439-441: quantitative results should be reported, for the sake of readability.

Lines 475-478: this should be included in a conclusion section of the manuscript, reporting also the weak points of the study.

Reviewer 4 Report

The topic of this paper regarding the robustness of physiological synchrony in wearable electrodermal activity and heart rate as a measure of attentional engagement to movie clips is interesting. The paper is generally well-written and seems to share comprehensive information on the topic with respect to the current literature. I have some remarks before I recommend the manuscript for publication:

 Minor comments:

The Authors should modify the number of participants in the part participants. It could be a bit confusing that in the abstract is N=29 but in the part participants N=30 (I know that it is explained in the text further, but this information could be mentioned in the part participants).

The Authors should supplement the part statistical analysis with all statistical methods used in the manuscript.

The Authors should check the Figures numbering, it is incorrect.  

The Authors should more explain the Figures in the legends of Figures.  

Round 2

Reviewer 2 Report

I am satisfied with the changes made. This paper maybe considered for publication now.

Author Response

-

Reviewer 3 Report

Manuscript ID: sensors-2158980

Title: Robustness of physiological synchrony in wearable electrodermal activity and heart rate as a measure of attentional engagement to movie clips

Recommendation: Major revision

Brief summary

This is the revised version of a manuscript I previously reviewed. The authors have considered most of my previous comments and in my opinion the quality of the paper has now been improved.

However, there are still some points needing to be revised by the authors.

According to my opinion, the advancements beyond the state of the art still need to be better explained in the Introduction section, also to prove the suitability of the manuscript for the publication in Sensors journal.

Some suggestions are provided in the next comments, which may help the authors in improving the quality of this paper.

Specific comments

Lines 139-141: the authors should report in a quantitative way the measurement accuracy of the used wearable sensors.

Lines 146-147: the levels of engagement attributed to the clips should be justified in a better way, in order to make the study reproducible also with clips, for example in a different language.

References: if possible, further references from Sensors journal should be reported, in order to better contextualize this study and also to better indicate its position within the topics of the journal.
